# Establishment of a CRISPR/Cas9-Mediated Efficient Knockout System of *Trichoderma hamatum* T21 and Pigment Synthesis PKS Gene Knockout

**DOI:** 10.3390/jof9050595

**Published:** 2023-05-19

**Authors:** Ning Luo, Zeyu Li, Jian Ling, Jianlong Zhao, Yan Li, Yuhong Yang, Zhenchuan Mao, Bingyan Xie, Huixia Li, Yang Jiao

**Affiliations:** 1Biocontrol Engineering Laboratory of Crop Diseases and Pests of Gansu Province, College of Plant Protection, Gansu Agricultural University, Lanzhou 730070, China; 18894313158@163.com; 2State Key Laboratory of Vegetable Biobreeding, Institute of Vegetables and Flower, Chinese Academy of Agricultural Sciences, Beijing 100081, China; l626984791@163.com (Z.L.); lingjian@caas.cn (J.L.); zhaojianlong@caas.cn (J.Z.); liyan05@caas.cn (Y.L.); yangyuhong@caas.cn (Y.Y.); maozhenchuan@caas.cn (Z.M.); xiebingyan@caas.cn (B.X.)

**Keywords:** *Trichoderma hamatum*, homologous recombination, CRISPR/Cas9, knockout efficiency

## Abstract

*Trichoderma hamatum* is a filamentous fungus that serves as a biological control agent for multiple phytopathogens and as an important resource promising for fungicides. However, the lack of adequate knockout technologies has hindered gene function and biocontrol mechanism research of this species. This study obtained a genome assembly of *T. hamatum* T21, with a 41.4 Mb genome sequence comprising 8170 genes. Based on genomic information, we established a CRISPR/Cas9 system with dual sgRNAs targets and dual screening markers. CRISPR/Cas9 plasmid and donor DNA recombinant plasmid were constructed for disruption of the *Thpyr4* and *Thpks1* genes. The result indicates the consistency between phenotypic characterization and molecular identification of the knockout strains. The knockout efficiencies of *Thpyr4* and *Thpks1* were 100% and 89.1%, respectively. Moreover, sequencing revealed fragment deletions between dual sgRNA target sites or GFP gene insertions presented in knockout strains. The situations were caused by different DNA repair mechanisms, nonhomologous end joining (NHEJ), and homologous recombination (HR). Overall, we have successfully constructed an efficient and convenient CRISPR/Cas9 system in *T. hamatum* for the first time, which has important scientific significance and application value for studies on functional genomics of *Trichoderma* and other filamentous fungi.

## 1. Introduction

*Trichoderma* is a class of filamentous fungi widely distributed in the plant rhizosphere ecosystem. It has biocontrol effects on many phytopathogens due to its competitive capacity, toxicity, mycoparasitic activity, antagonism, induced resistance, and growth promotion [1,2,3]. Some studies have reported that *Trichoderma* exhibited broad-spectrum inhibitory abilities against phytopathogens, including *Rhizoctonia solani* [4], *Fusarium verticillioides* [5], *F. oxysporum* [6], *Phytophthora infestans* [7], root-knot nematode [8,9] and cyst nematode [10]. *Trichoderma* also has obvious growth-promoting effects on tomatoes [11], cucumbers [12], and corn [13]. In addition, *Trichoderma* can improve seed germination rates and induce greater tolerance of the plant to stresses [14,15]. Secondary metabolites and spores are considered major factors that contribute to the biological control effects of *Trichoderma* in agricultural production [16]. At present, spore powder is the main component in commercial products of *Trichoderma*, but its control effect is susceptible depending on the environment in the field. Therefore, it is especially important to explore the mechanisms of *Trichoderma* in different environments to promote the efficient application of *Trichoderma*, such as induced systemic resistance (ISR). Secondary metabolites of *Trichoderma* are significant elicitors in ISR [16]. The volatile substances produced by *T. harzianum* and *T. asperellum* act as elicitors to stimulate the upregulation of *Arabidopsis thaliana*-induced resistance-related transcription factor *MYB72*, which activates the plant jasmonic acid pathway defense response [17]. Studholme et al. found that culture filtrates from *T. hamatum* GD12 can elicit a strong ISR response in rice against *Magnaporthae oryzae* [18]. However, the function identification, synthesis mechanism, and activation of silent gene clusters of secondary metabolites lack systematic studies in *T. hamatum*.

Gene knockout technology is an indispensable tool for research on mechanisms and secondary metabolites in filamentous fungi. The CRISPR/Cas9 system is based on RNA-mediated endonuclease to introduce DNA strand gaps at specific target sites in the genome, which can stimulate the host defense mechanisms to repair gaps and has already been implemented in multi-species [19,20]. With the help of the CRISPR/Cas9 system, the efficiency of homologous recombination has been improved, and the dependence on screening markers has been reduced. The CRISPR/Cas9 system has been successfully applied in various filamentous fungi in recent years, including *Ganoderma lucidum* [21], *Aspergillus niger* [22], *Penicillium chrysogenum* [23], and *Beauveria bassiana* [24]. For *Trichoderma* spp., the CRISPR/Cas9 system has only been successfully constructed in *T. reesei* and *T. harzianum* up to now [25,26]. In previous studies, the author could not obtain orotidine glycoside 5′-phosphate decarboxylase (*pyr4*) and pigment genes (*pks1*) knockout mutants by traditional homology recombination and split-marker methods [27] in *T. hamatum*. This may be attributable to the fact that the strain employs KU70 or KU80 as the dominant repair mechanism [28]. Therefore, the construction of an efficient CRISPR/Cas9 system is an essential step for more intensive studies of *T. hamatum* in the future.

Screening markers are one of the most important characteristics of fungus knockout research, including *pyr4* and pigment genes. The advantage of using *pyr4* and pigment genes as screening markers for fungal transformation is that knockout mutants and wild-type strains can be directly identified by phenotype. The *pyr4* encodes orotidine 5′- phosphate decarboxylase, which catalyzes the biosynthesis of uracil nucleotides and can convert 5-fluoro-orotic acid (5-FOA) into 5-fluorouracil (5-FU), interfering with RNA and DNA functions, leading to the death of the cell [29]. The growth of *pyr4* knockout mutants requires the supplementation of exogenous uracil or uridine and can be grown on a medium containing 5-FOA. Fungal spore pigments are mainly polyketides, mostly synthesized by polyketide synthases (PKSs). *Metarhizium robertsii* cannot produce heptaketide pigments after the *MrPks1* gene knockout. *MrPks1* is highly homologous with *Pks1* and *Pks2* in *T. reesei* whereas there are few relevant studies at present [30,31]. In this study, the disruption of *Thpyr4* and *Thpks1* genes demonstrated the successful use of the CRISPR/Cas9 system in *T. hamatum*. An efficient, rapid, and convenient fungal knockout system was established, which can lay the foundation for subsequent studies on the induced resistance mechanism of *T. hamatum*. Thus, this is an important technique with application value in *Trichoderma*-plant interactions, biocontrol mechanisms, and genetic manipulation.

## 2. Materials and Methods

### 2.1. Strain and Culture Conditions

*T. hamatum* T21 strain (CGMCC NO. 10923) was isolated, identified, and preserved by the Disease Group Laboratory of the Institute of Vegetable and Flower Research, Chinese Academy of Agricultural Sciences. *T. hamatum* T21 was cultured on potato dextrose agar (PDA: 200.0 g potato, 20.0 g glucose, 18.0 g agar to 1 L distilled water) medium at 28 °C. MOF medium (75.0 g mannitol, 15.0 g oat flour, 5.0 g yeast extract, 4.0 g glutamate, and 16.2 g MES to 1 L distilled water) was used as the non-pigment-producing culture to screen for pigment genes. Potato dextrose broth (PDB: 24.0 g of PDB powder per liter, BD-Pharmingen) medium was used for culturing *T. hamatum* T21 for fresh mycelium collection and DNA extraction. T-Top (0.5 g KCl, 0.5 g MgSO_4_·7H_2_O, 1.0 g KH_2_PO_4_, 2.0 g NaNO_3_, 200.0 g sucrose, 20.0 g glucose, and 10.0 g agar to 1 L distilled water) medium for protoplast regeneration. Luria Bertani (LB: 10.0 g peptone, 10.0 g NaCl, 5.0 g yeast extract, and 20.0 g agar to 1 L distilled water) medium containing kanamycin (50 μg/mL) or ampicillin (100 μg/mL) was used to culture *Escherichia coli* Trelief 5α (Tsingke, Beijing, China) at 37 °C for vector construction. Screening of resistance concentrations of *T. hamatum* T21 uses 5-FOA and Geneticin (G418).

### 2.2. Extraction of DNA and RNA

The total genomic DNA of *T. hamatum* T21 was extracted from mycelium grown on PDA for 7 days using the Fungal Genomic DNA Rapid Extraction Kit (Biotech, Shanghai, China). The total RNA was extracted from mycelium incubating on PDA and MOF after 7 days. RNA was extracted using FastPure Plant RNA (TianGen, Beijing, China) according to the manufacturer’s protocol. Subsequently, cDNA was reverse transcribed from RNA using the TIAN Script II RT kit (TianGen, Beijing, China).

### 2.3. Genome Assembly and Annotation

For single-molecule real-time (SMRT) sequencing, the polymerase reads were generated on the PacBio RSII platform, and the sequence quality was assessed according to the manufacturer’s protocol (Pacific Biosciences, Menlo Park, CA, USA). Finally, only subreads with a length of more than 500 bp and an RQ value higher than 0.75 were retained for future analysis. The Canu (v1.6) assembler was used for de novo assembly [32]. The genome was polished by three rounds of Pilon with the parameters-mindepth 10-changes-threads 4-fix bases using the Illumina short reads [33]. Gene structure, function annotation, and repeat annotation were carried out according to the operation manual [34].

### 2.4. Strain Identification

The genomic DNA of *T. hamatum* T21 was extracted as a template. The internal transcribed spacer (ITS) regions, translation elongation factor 1α (*tef1α*), and RNA polymeraseⅡ (*rpb2*) gene fragments were amplified via PCR using the primers ITS1/ITS4, tef1α-f/tef1α-r, and rpb2-f/rpb2-r [35,36]. Primer information is provided in Appendix A. The PCR program was as follows: 94 °C for 3 min, 34 cycles of 94 °C for 15 s, 56 °C for 15 s, and 72 °C for 15 s, with the final extension at 72 °C for 5 min. The phylogenetic trees were constructed by the maximum-likelihood method using MEGA 11.0 with default parameters and 1000 bootstrapping replicates.

### 2.5. Antibiotic Resistance Screening

*T. hamatum* T21 mycelium was inoculated into PDA medium containing different concentrations of 5-FOA at 0.5, 1.0, 2.0, and 3.0 mg/mL and cultured at 28 °C for 7 days to observe the growth status.

*T. hamatum* T21 mycelium was picked onto PDA medium containing different concentrations of G418 at 50.0, 100.0, 150.0, 200.0, and 250.0 μg/mL and cultured at 28 °C for 7 days to observe the growth status.

### 2.6. Evolutionary Analyses of Thpyr4 and Thpks1 Genes

Based on the function, we found all the sequences that have similarities to the *Thpyr4* and *Thpks1* genes in the NCBI database, and compared them with the *T. hamatum* T21 genomic sequences to determine the target gene sequences. The protein sequence of PKS1 (GenBank accession number XP_007823934.2) was used to search for homologous genes through an online tblastn search (http://www.ncbi.nlm.nih.gov/BLAST (accessed on 10 October 2022)) in the *T. hamatum* T21 genome [31]. The phylogenetic trees were constructed by the maximum likelihood method using MEGA 11.0 with 1000 bootstrap replicates. Prediction of functional domains of PKSs for each protein sequence using the PKS/NRPS analysis website (http://nrps.igs.umaryland.edu/ (accessed on 26 October 2022)) and the National Center Biotechnology Information (NCBI, Bethesda, MD, USA) database.

### 2.7. qRT-PCR Analysis

A 20 μL system was prepared using cDNA as a template with SYBR Premix Ex Taq II master mix (Takara, Shiga, Japan), and a BIO-RAD CFX96 (BIO-RAD, Hercules, CA, USA) instrument was used to analyze the relative expression levels. The primers are listed in Appendix A. Three putative candidate pigment genes were compared, and actin was selected as an internal reference for standardized measurement of gene expression. All quantitative real-time PCR reactions were repeated with at least three biological replicates and three technical replicates per reaction. The relative expression level was measured using the 2^−ΔΔCt^ method [37].

### 2.8. Construction of CRISPR/Cas9 Plasmid

CRISPR/Cas9 plasmid was obtained from the PUC-Cas9-neo-gRNA plasmid in our laboratory [38]. The target sequences of *Thpyr4* and *Thpks1* of the target genes were designed through the chop-chop (http://chopchop.cbu.uib.no/ (accessed on 5 November 2022)) website and two high-scoring candidate targets near the N-terminal region were selected [39,40]. The target RNAs of *Thpyr4* were 5′-TCCGTGCGAGGTGAAGACAT-3′ and 5′-ATGAGGAGACCTCGGTTCAG-3′. The target RNAs of pigment gene *Thpks1* were 5′-GGATTTGGAACCGGAATCTG-3′and 5′-TTAGAGCTAAACGTTGGCCA-3′. The seamless cloning primers are shown in Appendix A. Hammerhead (HH), sgRNA, and liver hepatic delta virus (HDV) ribozymes that sequentially form the sgRNA expression cassette were synthesized at Liuhe BGI Tech Solutions Co., Limited (Beijing, China). The linear fragments of the *gpda* promoter, *trpc* terminator, and sgRNA expression cassette were amplified by PCR and recovered by gel extraction and electrophoresis. The PUC-Cas9-neo-gRNA plasmid was digested with EcoRV and BglII (New England Biolabs, Ipswich, MA, USA) to recover the linear fragment. The Fast Pure Gel DNA Extraction Mini Kit (Vayzme, Nanjing, China) was employed to recover the PCR products for gel extraction. The four fragments (Pgpda, Ttrpc, sgRNA expression cassette and PUC-Cas9-neo) were assembled using Clon Express MultiS One Step Cloning Kit (Vayzme, Nanjing, China). The recombination products were transformed into *E. coli* Trelief 5α and cultured on an LB plate containing 100 μg/mL ampicillin at 37 °C for 12–16 h. Single colonies were picked for sequencing identification. Plasmid DNA was extracted using Plasmid Mini KitⅠ (OMEGA, Norcross, GA, USA).

### 2.9. Construction of Homologous Recombinant Plasmid

The PUC19 plasmids were digested with the restriction endoenzymes EcoRI and HindIII (New England Biolabs, Ipswich, MA, USA) as the vector backbone, and all primers were designed using a seamless cloning strategy (Appendix A). The upstream homology arm and downstream homology arm sequences were amplified using *T. hamatum* T21 genomic DNA as a template. The screening marker GFP fragment was amplified using the plasmid pch-sGFP as a template. The Clon Express MultiS One Step Cloning Kit (Vayzme, Nanjing, China) was used to ligate three fragments (upstream homology arm, GFP, and downstream homology) with overlapping sequences and insert them into the EcoRI and HindIII sites of the PUC19 backbone. *E. coli* Trelief 5α transformation and vector identification methods were the same as above. The up-f/down-r primers were used to linearize fragments as exogenous DNA using the recombinant plasmids as a template. The Fast Pure Gel DNA Extraction Mini Kit (Vayzme, Nanjing, China) was used to purify the PCR products for subsequent use.

### 2.10. Preparation of Protoplasts

*T. hamatum* T21 was incubated on PDA medium for 7 days. Conidia were collected by filtration and added to 200 mL PDB with shaking at 200 rpm, 28 °C for 14 h. The fresh mycelium was harvested by filtering through filter paper, and the PDB medium was removed by rinsing the mycelium with 0.7 mol/L NaCl 4–5 times. The mycelium was treated with 20 mg/mL driselase (Sigma, St. Louis, MO, USA) dissolved in STC buffer (58.3 g Sorbitol, 2.944 g CaCl_2_·2H_2_O, 0.6306 g Tris-HCl to 400 mL distilled water) at 28 °C, 140 rpm, for 3–4 h. The protoplasts were collected by filtration and centrifugation. The supernatant was discarded, and the concentration of protoplasts was adjusted to 1 × 10^6^ cells/mL with STC buffer. A 100 μL protoplast suspension was aspirated per transformation.

### 2.11. Co-Transformation of gRNA with Donor DNA

The transformation procedure was modified from the method of *Purpureocillium lilacinum* [38]. To a centrifuge tube containing 1 μmol of aurintricarboxylic acid, 2.5 μg of PUC-Cas9-gRNA plasmid and 2.5 μg of donor DNA repair template were added, the total volume was adjusted to 60 μL with TEC (0.3152 g Tris-HCl, 0.0585 g EDTA, 1.1762 g CaCl_2_·2H_2_O to 200 mL distilled water), and mixed well on ice for 20 min. Then, the mixture was centrifuged at 12,000 rpm for 2 min at 4 °C and the supernatant was removed by aspiration and mixed with 100 μL of protoplast suspension on ice for 20 min. To the above system, 160 μL of 60% PEG 4000 (120.0 g PEG 4000, 25.1 g MOPS to 200 mL distilled water) was added and mixed well at room temperature for 15 min. We added 1 mL of STC buffer, which was mixed well and centrifuged at 4000 rpm for 5 min at 4 °C. The supernatant was discarded, and the precipitation was resuspended in 200 μL of STC buffer. Finally, the total system was mixed with 25 mL T-Top medium and spread on 5 PDA plates homogeneously. For the *Thpyr4* gene treatment group, the T-Top medium contains 0.5 mg/mL uridine. The culture was incubated at 28 °C for 13–15 h. Then, 10 mL of screening T-TOP was added to each Petri dish. For the *Thpyr4* gene treatment group, T-Top contains 1.0 mg/mL 5-FOA and 0.5 mg/mL uridine. For the *Thpks1* treated group, T-Top contains 250 mg/mL G418. After incubation at 28 °C for 3–5 days, the growth status of the transformants was monitored daily and transferred to new PDA plates with corresponding antibiotics.

### 2.12. Identification of Transformant

The *Thpyr4* and *Thpks1* transformants were picked with toothpicks and placed into PDA Petri dishes with corresponding antibiotics. Pick the mycelium of transformants onto a slide and observe the green fluorescence under a fluorescence microscope (Olympus IX53, Tokyo, Japan). Then, *Thpyr4* and *Thpks1* transformants were collected after 15–20 h of incubation at 28 °C and 200 rpm for extraction of the genomic DNA. Positive transformants were identified by PCR, and the information on primers is listed in Appendix A.

### 2.13. RT-PCR Identification for Cas9 Expression

Three Δ*Thpyr4* mutants were randomly selected to extract RNA and make cDNA. A pair of primers were designed for the Cas9 amplified fragment of 1113 bp for identification (Appendix A), and cDNA was used as a template for PCR. The PCR program was as follows: 94 °C for 3 min, 34 cycles of 94 °C for 15 s, 56 °C for 15 s, and 72 °C for 30 s, and a final step at 72 °C for 5 min.

### 2.14. Knockout Efficiency Statistics

The *Thpks1* knockout transformants were selected by G418 antibiotic and GFP markers cultured in 24-well cell culture plates, and knockdown efficiency was calculated in combination with phenotypic and molecular identification.

The calculations are presented in the following formula:Knockdown efficiency (%)=(number of ΔThpks1/total number of transformers)×100%

## 3. Results

### 3.1. Strain Identification and Genome Annotation

The PCR products were approximately 600, 321, and 948 bp for the ITS, *tef1α*, and *rpb2*, respectively (Appendix A). The result of sequencing and the BLAST search showed the ITS, *tef1α*, and *rpb2* shared a sequence identity of over 99% with *T. hamatum.* For the phylogenetic tree analysis, the dataset comprised 16 taxa, including *Escovopsis clavata* as the outgroup (Appendix A). Based on the ITS, *tef1α*, and *rpb2* sequences, the T21 strain and *T. hamatum* were clustered with each other, with 100% bootstrap support in three phylogenetic trees (Appendix A).

*T. hamatum* T21 genome data has been deposited at the National Genomics Data Center under accession number GWHBWEF00000000. The *T. hamatum* T21 genome size was estimated to be 41.4 Mb based on the k-mer statistics from an Illumina paired-end library with an insert size of 380 bp. We obtained 3.84 Gb (~92×) of long reads using the PacBio SMART platform for *T. hamatum* T21. The PacBio subreads were assembled into contigs, and a total of 251 contigs were obtained for *T. hamatum* T21, with an N50 of 2.04 Mb (Figure 1). To assess the assembly accuracy, we remapped the raw reads of the paired-end library to the assembled *T. hamatum* T21 genome. The reads covered 98.64% of the genome, with a 94.25% mapping rate and 78× average sequence depth, which implied that the current *T. hamatum* T21 assembly covered almost all unique genomic regions. The genome of *T. hamatum* T21 contained 16.17 Mb repeat sequences, accounting for 38.45% of the genome size. A total of 8170 protein-coding genes were predicted for *T. hamatum* T21. BUSCO assessment of the *T. hamatum* T21 genome showed that 1325 (94.85%) of the gene models were complete (Appendix A), suggesting that the assemblies included most of the *T. hamatum* T21 gene space.

### 3.2. Screening of Resistance Concentration of T. hamatum T21 to 5-FOA and G418

The results of the responses of *T. hamatum* T21 to antibiotics are shown in Figure 2. *T. hamatum* T21 grew normally on PDA plates without 5-FOA, and growth was inhibited on PDA plates with 5-FOA concentrations of 0.5, 1.0, 2.0, and 3.0 mg/mL. The growth inhibition effect of 5-FOA on *T. hamatum* T21 was limited to a concentration of 0.5 mg/mL, so 1.0 mg/mL was used as the screening concentration (Figure 2A).

*T. hamatum* T21 grew normally on PDA plates without G418, while growth was inhibited on PDA plates with G418 concentrations of 200 and 250 μg/mL. The growth of colonies was less inhibited at a concentration of 200 μg/mL. Therefore, 250 μg/mL was used as the G418 screening concentration for transformation in this project (Figure 2B).

### 3.3. Thpks1 Candidate Genes and qRT-PCR Analysis

By comparing the pigment genes of *M. robergii*, *g3839*, *g4143*, and *g6889* in *T. hamatum* T21 were selected as candidate genes with high homology. *T. hamatum* T21 produced green pigment on PDA, but could not produce it on MOF medium (Appendix A). Thus, the expression levels of the three candidate genes were measured by qRT-PCR on the two mediums. These results showed that the expression of *g4143* was significantly upregulated 2216-fold in the PDA medium compared to the MOF medium, and we named *g4143* as *Thpks1* by sequence analysis (Appendix A). The open reading frame (ORF) of *Thpks1* consists of 2454 bp and encodes an 817 amino acid protein. Phylogenetic analyses of the protein sequences of *Thpks1* and other PKS suggested that Thpks1 clustered with the conidial pigment biosynthesis polyketide synthase of *M. robertsii* ARSEF23 (Genebank accession E9F646.2). *Thpks1* encoded a highly reducing PKS (NR-PKS) containing ketosynthase (KS), acyltransferase (AT), product template domain (PT), acyl carrier protein (ACP), and thioesterase (Te) domains, and was homologous to other known functions of NR-PKS (Figure 3), such as PKS-melA (Genebank accession A0A0A2KT65.1) in *P. expansum,* which encodes a PKS responsible for yellow pigment formation, and PKSA (Genebank accession M2XHZ5.1) in *Dothistroma septosporum*, which encodes dothistromin with structural similarity to the aflatoxin precursor versicolorin B [41,42].

### 3.4. Thpyr4 and Thpks1 Gene Knockout Vector Construction

BLAST analysis of protein sequences from the NCBI database indicates that the orotidine 5′-phosphate decarboxylase gene of *T. gamsii* shares a higher sequence identity with *g1974* of the *T. hamatum* T21 genome (96.83%) (Appendix A). Thus, *g1974* was determined to be a target gene and named *Thpyr4*.

In the previous phase, the target gene could not be knocked out by homologous recombination or split-tagging using conventional knockout methods. CRISPR/Cas9 and homologous recombinant vectors using *Thpyr4* and *Thpks1* genes as targets were constructed to establish an efficient, convenient gene knockout system. The CRISPR/Cas9 system was optimized based on the previous study in our laboratory [38]. The fragments of Pgpda, sgRNA expression cassette, and Ttrpc were inserted into the EcoRV and BglII sites of the PUC-Cas9-neo plasmid to constitute a CRISPR/Cas9 knockout cassette (Figure 4). The above vector has only one effective screening marker, G418, and Cas9 can cleave the gene target to form a double-strand break. The cleavage sites can be repaired by HR or NHEJ. Therefore, a homologous recombinant plasmid was constructed in which the GFP was a selection marker for the alternative target gene (Figure 4).

### 3.5. Thpyr4 Knockout and Identification

The Thpyr4-up-f/Thpyr4-down-r primers were selected to linearize fragments using homologous recombinant plasmids as templates and mixed with CRISPR/Cas9 plasmids for PEG-mediated fungal transformation. The transformant mycelium showed green fluorescence under fluorescence microscopy, indicating the successful transformation of GFP (Figure 5A,B). The transformants were identified with Thpyr4-f/Thpyr4-r and Thpks1-f/Thpks1-r, followed by genomic DNA extraction. Thpyr4-f/Thpyr4-r amplified 0 bp (lanes 3 and 4) and 603 bp (lanes 1, 2, and 5) from the knockout strain, but 872 bp from T21 wild type (Figure 5D). The Thpks1-f/Thpks1-r primers amplified an 816 bp (lanes 8 to 12) fragment from the knockout strain, and T21 wild type suggested the high quality of the DNA template (Figure 5D).

Sequencing and alignment results indicated that the transformants 1, 2, and 5 (lanes 1, 2, and 5, Figure 5D) were successfully cut by Cas9 between the dual sgRNA targets (Appendix A). Additionally, for transformants 3 and 4 (lanes 3 and 4, Figure 5D), *Thpyr4* was partly replaced by an inserted GFP marker at the shear site. The Δ*Thpyr4* strains were unable to grow on PDA medium without extra uridine or uracil. The Δ*Thpyr4* can grow on PDA medium with uridine or uridine and 5-FOA. Additionally, it grew fastest on PDA medium containing only uridine. The consistency between the phenotype and molecular identification of Δ*Thpyr4* indicates successful knockout of the *Thpyr4* gene (Figure 5C). The Δ*Thpyr4* mutants grew well after six subcultures. After subculture, the fluorescence intensity of the mutants showed no change, indicating that the GFP gene was stably inherited in *T. hamatum* T21. Furthermore, RT-cas9-f and RT-cas9-r amplified 1113 bp DNA fragment from the Δ*Thpyr4* mutants but 0 bp from T21 wild type, which indicated that the Cas9 protein was successfully expressed in Δ*Thpyr4* (Figure 5E). Thus, through the CRISPR/Cas9 gene knockout technique, uridine/uracil auxotrophic strains of *T. hamatum* T21 were obtained.

### 3.6. Thpks1 Knockout and Identification

Homologous recombinant plasmids served as templates, and Thpks1-up-f/Thpks1-down-r primers were used to amplify homologous recombinant fragments. After PCR purification, the linearized fragments were mixed with CRISPR/Cas9 plasmids for PEG-mediated fungal transformation. The transformant mycelium and conidia appeared green under fluorescence microscopy, which indicated the successful insertion of GFP (Figure 6A,B). Five transformants were randomly selected, and genomic DNA was extracted. The Thpks1-f/Thpks1-r amplified 0 bp (lanes 1 to 4) and a 512 bp (lane 5) DNA fragment from the knockout strain but 816 bp (lane 6) from T21 wild type (Figure 6E). The Thpyr4-f/Thpyr4-r primers amplify 872 bp (lanes 8 to 13) DNA fragment from the knockout strain and T21 wild type, indicating the high quality of the DNA template (Figure 6E).

The sequencing and alignment results indicated that the *Thpks1* of transformant 5 (lane 5, Figure 6E) was successfully cut off by the Cas9 protein between the dual targets (Appendix A). The Δ*Thpks1* mutants showed white colonies absent of green pigment, which is a significant phenotypic difference compared to the wild type and consistent with molecular identification (Figure 6C,D). Transformant genomic DNA with green fluorescence was extracted. After extending the elongation time of PCR, amplified with Thpks1-f/Thpks1-r primers, the products were 3982 bp from the knockout strain (lanes 1 to 5) but 816 bp (lane 6) from the T21 wild type (Figure 6F). The above result indicated that *Thpks1* was partly replaced at the shear sites by the insertion of the GFP marker gene.

### 3.7. Knockout Efficiency Statistics

After screening by G418, 242 transformants were acquired, 175 of which had green fluorescence observed under the fluorescence microscope. All the transformants with green fluorescence were transferred to 24-well cell plates for phenotypic observations and molecular identification. Results are presented in Figure 7. Thpks1-f/Thpks1-r amplified an 816 bp (lanes 14, 18, 19, and 21) DNA fragment from the transformer of the green phenotype aligned to T21 wild type. The white phenotype transformers were successfully knocked out of the pigment gene *Thpks1*, which corresponds to the molecular identification results. Additionally, the Thpks1-f/Thpks1-r amplified a 512 bp (lanes 4, 10, 11, and 17) fragment from knockout mutants (Figure 7B), and sequence alignment of lane 4 and lane 11 indicated that this phenomenon was due to NHEJ between the dual targets (Appendix A). Based on phenotype and molecular identification, 156 transformers were successfully knocked out with *Thpks1*, of which 27 knockout mutants were due to Cas9 nuclease shearing, and 129 knockout mutants had successful insertion of the GFP at the dual target site (Appendix A). Therefore, the simultaneous transfer of CRISPR/Cas9 plasmids and the homologous recombinant fragment was able to significantly improve the knockdown efficiency by 89.1%.

## 4. Discussion

*Trichoderma* spp. is not only a parasitic fungus against many plant pathogens but also an important biocontrol resource for developing microbial fungicides [43]. The application and mechanistic studies of *Trichoderma* spp. biocontrol have become increasingly important in plant disease control [44,45]. Recent studies showed that *T. hamatum* has high biocontrol activity against a variety of plant pathogens, as well as stimulating plant growth and defense responses [18,46]. Therefore, *T. hamatum* has been used in biological preparations for biological control. In this way, gene knockout is an important means for the improvement of *T. hamatum* strains and biocontrol studies. In our previous study, neither homologous recombination nor split marker methods could successfully knock out the genes of *T. hamatum* T21. For this, we have successfully constructed an efficient, convenient, and versatile CRISPR/Cas9 system in *T. hamatum* for the first time, which is critical to investigating the mechanism of biocontrol. In this system, the Cas9 protein will induce DSBs in the target gene. DSBs can be repaired by NHEJ or HR to create genomic alterations, gene insertions, and gene knockouts (Figure 4). This is the main reason for improving knockout efficiency in the *T. hamatum* T21 genome. This technology also lays the foundation for further studies on genetic improvement and gene function, not only in *T. hamatum* but also in other *Trichoderma* species.

The genetic background of filamentous fungi is complicated. The homologous recombination technique has been gradually replaced by CRISPR/Cas9 technology due to its low efficiency, high off-target rates, and heavy workload [47]. For a wide variety of fungi, the paucity of screening markers is an extreme limitation to the knockout of multiple genes. GFP is a fluorescent tag that has been applied in many ways. The recombinant expression of GFP that was inserted into the target expression vector made it possible to evaluate the effect of the vector protein expression visually. After the transformation of GFP, bright green fluorescence was observed in the *Trichoderma* mycelium under 488 nm excitation light. Thus, the exogenous donor DNA plasmid PUC19-eGFP was co-transformed with the CRISPR/Cas9 plasmid to increase knockdown efficiency in this work, and its knockout rate of the *Thpyr4* gene reached 100%. In this study, the simultaneous transformation of two plasmids (CRISPR/Cas9 and PUC19-GFP) could significantly improve the knockout efficiency (89.1%). 17.3% of mutants belonged to Cas9 nuclease sheared between the dual sgRNA, indicating that GFP inserts randomly in the *T. hamatum* T21 genome. The CRISPR/Cas9 system illustrated feasibility, practicability, and stability combined with the phenotypic characterization and molecular identification of Δ*Thpyr4* and Δ*Thpks1* mutants. In addition, with the advantage of counterselection using 5-FOA, *Thpyr4* could be utilized as a bidirectional selection marker to resolve the antibiotic screening marker deficiency and achieve continuous multigene knockout in the fungus [48].

Compared with conventional knockout methods, the CRISPR/Cas9 system had the advantages of simple construction, a specific target, and high knockout efficiency. The primary factors affecting knockout efficiency included fungal individual characteristics, promoter strength, transformation approaches, the length of the homology arms, the sgRNA target sequence C/G content, and gRNA expression [49,50,51,52,53,54,55,56,57]. Knockout efficiency varied with different genes and targets using the CRISPR/Cas9 system in the same strain. Katayama et al. chose the U6 promoter in the CRISPR/Cas9 system with an editing efficiency of 10–20%, 10%, and 100% for *WA*, *pyrG*, and *yA* genes, respectively, in *A. oryzae* [58]. Gabriel et al. designed two different sgRNA targets for the same gene in *Thermoascus aurantiacus* and showed that the knockout efficiency of the two targets was 10% and 35%, respectively [59]. The number of target genes also affected the knockout efficiency. In *T. reesei*, the double knockout rate was 43%, and only 4.2% of the simultaneous knockout efficiency of triple genes [25]. The dual sgRNA system has been used to delete target DNA fragments in *G. lucidum*; the knockdown efficiency of *URA3* was 36.7%, which was significantly different from our study [60]. In our study, the length of the homologous arm was designed to be 2000 bp, and each gene was targeted with two specific gRNAs, which significantly increased the efficiency of homologous recombination. There was a significant difference in the number of Δ*Thpyr4* and Δ*Thpks1*, with 13 and 242, respectively, which may be due to the toxic effects of 5-FOA on protoplasts.

The fungal conidial pigments are essential components for the formation of the fungal cell wall, which plays a very important role during the growth and development of fungi [61]. Conidia pigments affect fungal conidia development, UV tolerance, and pathogenicity. Carotenoid pigments produced in fungi protect against oxidative stress and visible light or UV irradiation [62]. The pigment aspmelanin in *A. terreus* is resistant to UV irradiation, and asparasone in *A. flavus* is crucial for sclerotial survival [63]. The melanin in *Pestalotiopsis microspora* has a negative effect on conidia but is also important for maintaining cell integrity and viability [64]. The above studies enrich our understanding of the importance of fungal pigments. In our work, based on the bioinformatics analysis of known fungal conidia pigment synthesis genes, we predicted the homologous gene *Thpks1* for conidia pigment synthesis genes in *T. hamatum*. Thpks1 showed homology and conservation with other proteins related to pigment syntheses, such as the *alb1* protein in *A. fumigatus*, the *pksA* protein in *D. septosporum*, the *melA* protein in *P. expansum*, and the pks protein in *M. robertsii* (Figure 3). The *Thpks1* knockout mutants were obtained by the CRISPR/Cas9 system and confirmed that *Thpks1* was a key gene of the conidia green pigment synthesis in *T. hamatum*. Herein, we report *Thpks1* as the major green pigment synthesis gene of *T. hamatum* for the first time and provide a screening marker for genetic manipulation of *Trichoderma*. Combining phenotypic and molecular identification demonstrated the feasibility and accuracy of the CRISPR/Cas9 system. For the moment, the biosynthesis of green pigments is more complex in *Trichoderma*, and the biosynthetic pathway and regulatory mechanisms require further investigation in the future.

## 5. Conclusions

In this study, we obtained the genome assembly for *T. hamatum* T21. Based on the CRISPR/Cas9 system, a dual sgRNA efficient knockout method was established in *T. hamatum* T21 successfully. The adoption of dual sgRNA and dual screening marker strategies has notable strengths in gene knockout of *T. hamatum*, having advantages of simplicity, convenience, and high efficiency. This strategy performed quickly and facilitated the knockout of individual genes in the *T. hamatum* genome. The establishment of this genome editing system will provide an efficient tool to intensively investigate the mechanism of induced resistance and elucidate the secondary metabolite synthesis pathways in *T. hamatum*. Furthermore, our results will bring breakthroughs for further investigations on the functional analysis of biocontrol genes, as well as the elucidation of molecular mechanisms behind the agricultural applications of filamentous fungi.

## Figures and Tables

**Figure 1 jof-09-00595-f001:**
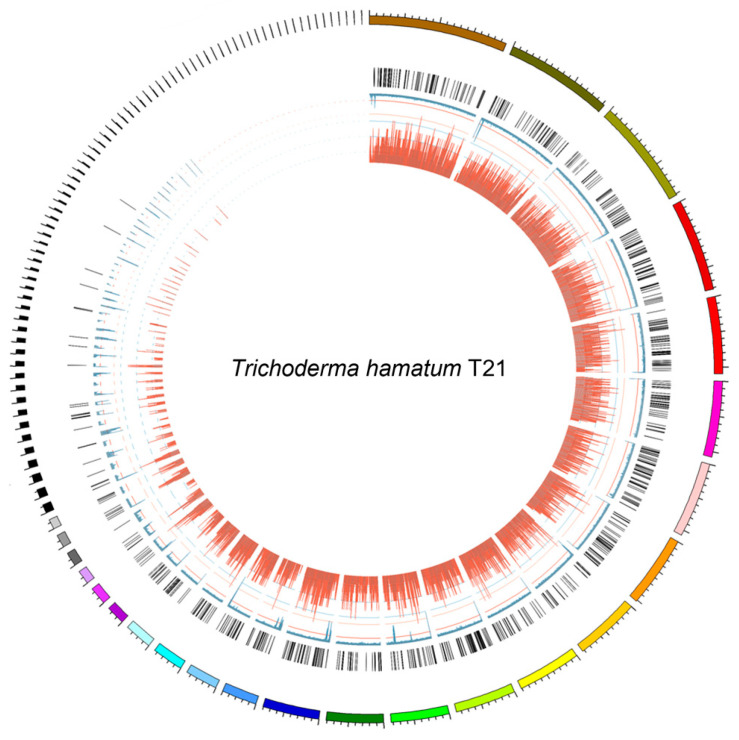
Genome feature of *T. hamatum* T21. The outermost circle is the contigs. The bar charts from outside to inside in turn show secreted proteins (orange), the density of repetitive sequence (blue), and gene density (dark red).

**Figure 2 jof-09-00595-f002:**
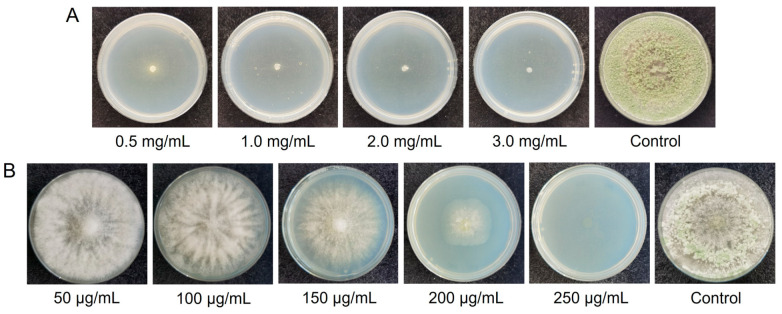
Response of T. hamatum T21 to two antibiotics at different concentrations. The antibiotics are 5-fluoroorotic acid (**A**) and G418 (**B**). The concentration of antibiotics is indicated below the picture.

**Figure 3 jof-09-00595-f003:**
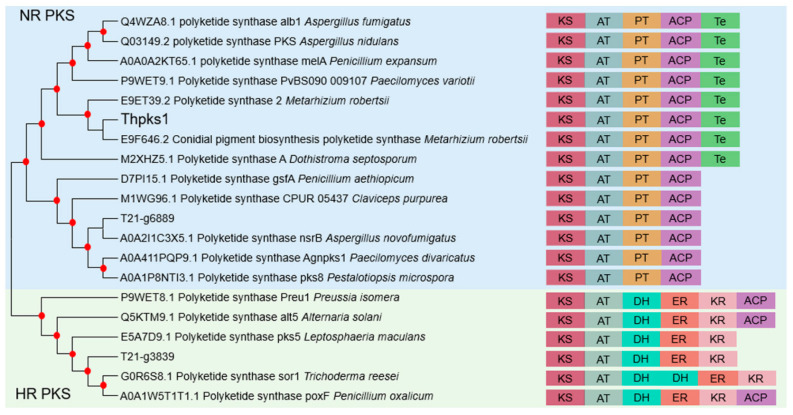
Phylogenetic analysis of the PKS domain. Branch nodes with greater than 70% support from 1000 bootstrapped pseudo-replicates are indicated with red dots in the phylogenetic tree. NR-PKS indicates nonreducing PKS, and HR-PKS indicates highly reducing PKS. KS, ketosynthase domain; AT, acyltransferase domain; PT, product template domain; DH, dehydratase domain; ER, enoylreductase domain; KR, β-ketoacylreductase domain; ACP, acyl carrier protein; Te, thioesterase.

**Figure 4 jof-09-00595-f004:**
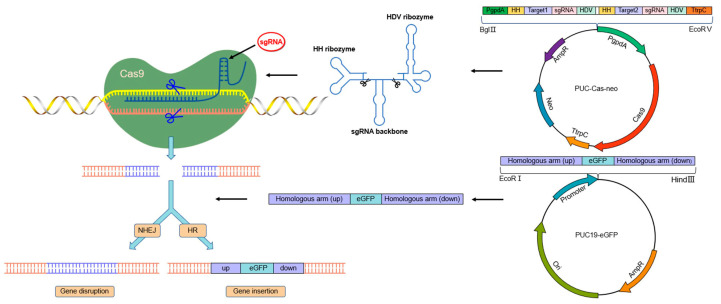
Schematic representation of knockout genes by a CRISPR/Cas9-mediated homologous recombination strategy. CRISPR/Cas9 system releases generate functional sgRNAs and direct Cas9 cleavage to complementary sites in the genome. Primary sgRNA transcripts contain HDV and HH ribozymes, which are self-cleaved to release functional sgRNAs that direct Cas9 cleavage to complementary sites in the genome. The black scissors indicate the cleavage sites. The target gene cut site is cut by a gRNA/Cas9 complex, and a double-strand break will be formed, activating cellular repair mechanisms. Cas9 cutting sites are indicated with blue scissors. Homologous recombination (HR) and nonhomologous end-joining (NHEJ) are two major pathways of DNA double-strand break repair, leading to loss of gene capacity and achieving gene knockout.

**Figure 5 jof-09-00595-f005:**
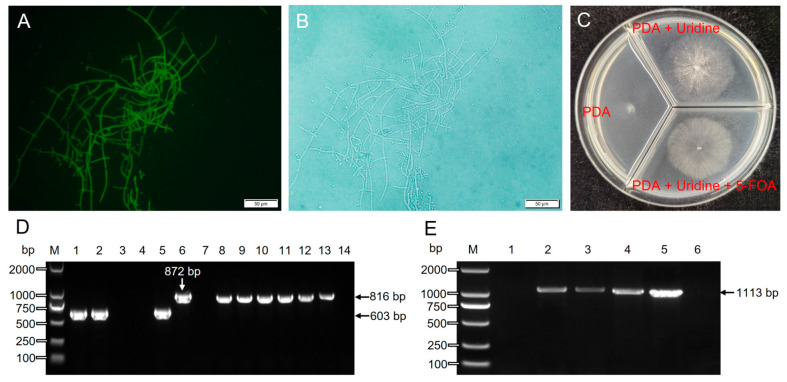
Phenotypes and molecular identification of Δ*Thpyr4* mutants and analysis of Cas9 protein expression. (**A**,**B**) Fluorescent images and bright field of the *Thpyr4* knockout strain. (**C**) Growth phenotypes of mutants on medium containing uridine, uridine + 5-fluoroorotic acid. (**D**) Molecular identification of the Δ*Thpyr4* mutant. M, Marker. Lanes 1 to 5, amplification of five knockout strains. Lane 6, positive control, amplification of the T21 wild-type strain. Lane 7, negative control, amplification of water. Lanes 8 to 14, template validation, templates same with 1–7 amplified by Thpks1-f and Thpks1-r. (**E**) RT-PCR analysis confirming Cas9 expression in the Δ*Thpyr4* mutants. M, Marker. Lane 1, T21 wild type. Lanes 2 to 4, Δ*Thpyr4* mutants. Lane 5, positive control, amplification of the CRISPR/Cas9 plasmid. Lane 6, negative control.

**Figure 6 jof-09-00595-f006:**
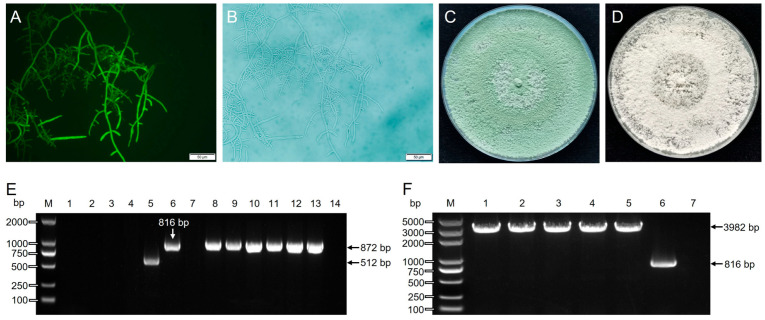
Phenotypic characterization and molecular identification of the Δ*Thpks1* strain. (**A**,**B**) Microscopy fluorescent and bright field images of the Δ*Thpks1* strain. (**C**,**D**) Phenotypic characteristics of T21 wild type and Δ*Thpks1* strains under the same culture conditions. (**E**) Molecular identification of the Δ*Thpks1* strain. M, Marker. Lanes 1 to 5, amplification of five knockout strains. Lane 6, positive control, amplification of the T21 wild type. Lane 7, negative control, amplification of water. Lanes 8 to 14, template validation, templates same with 1–7 amplified by Thpyr4-f and Thpyr4-r. (**F**) Verification of homologous recombination of the GFP marker gene at cleavage sites. M, Marker. Lanes 1 to 7 are amplified by Thpks1-f and Thpks1-r. Lanes 1 to 5, amplification of five knockout strains. Lane 6, positive control, amplification of the T21 wild type. Lane 7, negative control, amplification of water.

**Figure 7 jof-09-00595-f007:**
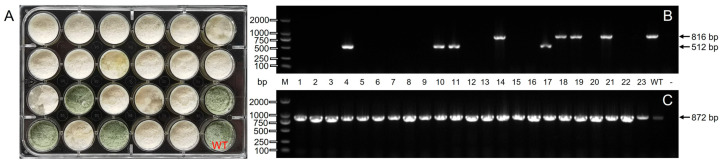
Statistical knockout efficiency combined with phenotypic and molecular identification of Δ*Thpks1* strain. (**A**) The phenotype of transformants on a 24-well culture plate. WT represents the T21 wild-type strain and served as a control. (**B**) PCR analyses of T21 wild type and *Thpks1* knockout strains. M, Marker. Lanes 1 to 23, amplification of 23 transformants. Lane WT represents the T21 wild-type strain and served as a positive control. Lane—represents negative controls. (**C**) Template validation, all templates amplified by Thpyr4-f and Thpyr4-r.

## Data Availability

Not applicable.

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
