# Peer review of "Establishment of a CRISPR/Cas9-Mediated Efficient Knockout System of Trichoderma hamatum T21 and Pigment Synthesis PKS Gene Knockout"

_jof, 2023, doi:10.3390/jof9050595_

Round 1
Reviewer 1 Report
I have just some minor revisions:
Article
Establishment of CRISPR/Cas9-mediated efficient knockout system of Trichoderma hamatum T21 and pigment synthesis PKS gene knockout
Abstract
line 12 - "and an important resource for the development of fungicides". It will fit better to write "and AS an important resource for the development of fungicides"
Introduction
line 76 - "Metarhizium Roberts". You mean Metarhizium robertsii
Results
line 264 - Figure word is duplicated
Figure 2 legend - T. hamatum must be in italic
English is OK.
Author Response
Response to Reviewer 1 Comments
Dear Reviewer,
Thank you for your comments concerning our manuscript entitled “Establishment of CRISPR/Cas9-mediated efficient knockout system of Trichoderma hamatum T21 and pigment synthesis PKS gene knockout” (jof-2378322). Those comments are valuable and very helpful. We have read through comments carefully and have made corrections. The responses to your comments are presented following.
Point 1:
Abstract
line 12 - "and an important resource for the development of fungicides". It will fit better to write "and AS an important resource for the development of fungicides"
Response 1: We sincerely thank the reviewer for careful reading. As suggested by the reviewer, we have corrected the “and an important resource for the development of fungicides” into “and AS an important resource for the development of fungicides.
Point 2:
Introduction
line 76 - "Metarhizium Roberts". You mean Metarhizium robertsii
Response 2: We were sorry for our careless mistakes. In our resubmitted manuscript, the typo is revised. Thank you for your reminder.
Point 3:
Results
line 264 - Figure word is duplicated
Figure 2 legend - T. hamatum must be in italic.
Response 3: We have corrected it according to your comments.
Special thanks to you for your good comments.
Reviewer 2 Report
In the present work, the authors established an elegant CRISPR/Cas9 system with dual target sgRNAs and dual selection markers. The target genes were Thpyr4 and Thpks1 genes. The paper is well written and with a good discussion, however, following are some major and minor observations for the authors to consider:
Major:
I consider the strategy of constructing the CRISPR/Cas9 and homologous recombination vectors to be one of the most interesting parts of the paper and one that will serve as a model for future work by other authors, so,
1. Consider describing the map of the vectors in detail in the Materials and Methods section as a figure and, if you prefer, in simplified form in Figure 4 of the Results section.
2. In the Materials and Methods session or in the Discussion session, consider describing a paragraph with the bottleneck points, i.e., what are the main difficulties encountered by the authors for the development of the system.
Minor:
1. In the legends of figures 2 and 3, (line 274) T. hamatum and the name of the PKS gene (line 296) please put in italics and revise the text accordingly.
2. In the figure legends with the amplifications of PCR fragments (figures 5, 6 and 7), please indicate the size of the expected amplicons.
Author Response
Response to Reviewer 2 Comments
Dear Reviewer,
Thank you for your comments concerning our manuscript entitled “Establishment of CRISPR/Cas9-mediated efficient knockout system of Trichoderma hamatum T21 and pigment synthesis PKS gene knockout” (jof-2378322). Those comments are valuable and very helpful. We have read through comments carefully and have made corrections. The responses to your comments are presented following.
Point 1: Consider describing the map of the vectors in detail in the Materials and Methods section as a figure and, if you prefer, in simplified form in Figure 4 of the Results section.
Response 1: Thank you for your nice comments on our article. After consultation with all authors, we describe this in more detail in the Materials and Methods section. In addition, we still keep the original Figure 4, not adopting your comments to revise this part. We did this for the three following reasons. First of all, descriptions of the main methods used in this study are included in Figure 4. The process of constructing plasmids is described in detail in the Materials and Methods section, and the order of the fragments has been shown in detail in Figure 4. Secondly, the process of constructing the plasmid, the CRISPR/Cas9 mechanism, and the predicted results are included in Figure 4. The method of constructing the vector and the results obtained are clearly expressed. A schematic diagram of the CRISPR/Cas9 mechanism and results illustrating the principle of an efficient CRISPR/Cas9-mediated knockdown system is also included in Figure 4. Based on genomic information, we established a CRISPR/Cas9 system with dual sgRNAs targeting and dual screening markers. The principle and workflow of CRISPR/Cas9 plasmid and homologous recombinant plasmid PUC-eGFP are clearly illustrated in Figure 4. This will make the article hierarchical and fully reflect the main idea of the article. Finally, Figure 4 is a label for this article which supplied reasonable details of the article's subjects. This makes the main idea of the article clearer, more reasonable layout, and is more completely structured. The reader is more legible and easier to understand the topic of the article.
Point 2: In the Materials and Methods session or in the Discussion session, consider describing a paragraph with the bottleneck points, i.e., what are the main difficulties encountered by the authors for the development of the system.
Response 2: We sincerely appreciate the valuable comments. We tried to answer this question in two ways. First, a bottleneck point was encountered before developing CRISPR/Cas9 system. The author could not obtain Thpyr4 and Thpks1 knockout mutants by traditional homology recombination and split-marker methods in T. hamatum T21. To solve the dilemma, we performed several experiments and change the experimental schema, including the selection of screening markers (hygromycin B and G418), changing the length of the homologous arm (1000 bp, 1500 bp, 2000 bp, and 3000 bp), increasing the number of identified transformers, and increasing the concentration of homologous recombinant fragments. However, the multiple attempts we made have not yet been successful. This may be attributable to the fact that the strain employs KU70 or KU80 as the dominant repair mechanism. Therefore, the construction of an efficient CRISPR/Cas9 system is an essential step for more intensive studies of T. hamatum. We have added a narrative for the point in the Introduction section and Discussion section. The second point is the analysis of the knockout efficiency of the CRISPR/Cas9 system. Throughout this experiment, we came across notable findings. After screening by G418, 242 transformants were acquired, and 175 of them showed green fluorescence under the fluorescence microscope. Based on phenotype and molecular identification, 156 transformers were successfully knockout with Thpks1, of which 27 knockout mutants were due to Cas9 nuclease shearing. In other words, these 27 mutants generated by Cas9 protein shearing have green fluorescence, but the insertion into the target locus of GFP by homologous recombination did not occur. Therefore, we consulted the kinds of literature, this is due to the random insertion of GFP into the genome. However, 129 knockout mutants with successful insertion of the GFP at the dual target site, achieved homologous recombination. The simultaneous transformation of two plasmids (CRISPR/Cas9 and PUC19-eGFP) could significantly improve the knockout efficiency (89.1%). 17.3% of mutants belonged to Cas9 nuclease sheared between the dual sgRNA, indicating that GFP inserts randomly in the T. hamatum T21 genome. Compared with previous studies, the efficiency of homologous recombination increased strikingly with CRISPR/Cas9-mediated editing, with a knockout efficiency of 73.7%. What’s more, the simultaneous construction of homologous recombinant fragments and CRISPR/Cas9 into the same vector is the subject of our future study. This part of the analysis is described in the Discussion section.
Point 3: In the legends of figures 2 and 3, (line 274) T. hamatum and the name of the PKS gene (line 296) please put in italics and revise the text accordingly.
Response 3: Thank you for this suggestion. We have made corrections in the legends of Figure 2 based on your comments. In Figure 3, PKS represents polyketide synthase and the sequences used for building the tree were protein sequences. Therefore, the PKS here is orthomorphic and we have modified the figure legend of Figure 3 according to your comments.
Point 4: In the figure legends with the amplifications of PCR fragments (figures 5, 6 and 7), please indicate the size of the expected amplicons.
Response 4: According to your suggestions, we have indicated the size of the expected amplicons in Figures 5, 6 and 7.
Thank you again for your suggestion. I hope to learn more knowledge from you.
Reviewer 3 Report
The authors have performed whole genome sequencing and analysis for Trichoderma hamatum T21 and successfully established CRISPR/Cas9-mediated genome editing system. This is a well-written manuscript and can be considered for publication in the Journal of Fungi with some changes. Here, providing my comments to improve the manuscript.
1. The Authors need to recheck the writing of the gene name in the title - PKS. I think it should be in italics.
2. Correct how to write a term - CRISPR/Cas9 or CRISPR-Cas9 (Line 57 and several other instances). Following a uniform way of writing the text throughout the draft would be ideal.
3. Line 158-170 – sgRNA (in fact, spacer) sequences generally are without the PAM nucleotides, but authors have included them in spacers. Also, the term- sgRNA or gRNA sequence is named for the combination of both the spacer (crRNA) and scaffold (tracrRNA) region (refer to this - https://doi.org/10.1007/978-981-16-9720-3_10). Revise the text to include the correct information.
4. Figure 2- Labels added inside the panel can be better reorganized.
5. Figure 4- Cas9, not cas9; HR arm labeling in the gene disruption- up and down or up and up?
6. As mentioned by the authors, in previous studies, the knockout experiments for the pyr4 and pks1 genes were unsuccessful, possibly due to different DNA repair modes in other strains. What is the author's opinion about succeeding the KO of these genes in the tested strain? This aspect should be discussed more clearly in the discussion section.
7. The authors should differentiate the data and their interpretation of HR and gene knockouts generated for targeted genes.
This is a well-written manuscript.
Author Response
Response to Reviewer 3 Comments
Thank you for your comments concerning our manuscript entitled “Establishment of CRISPR/Cas9-mediated efficient knockout system of Trichoderma hamatum T21 and pigment synthesis PKS gene knockout” (jof-2378322). Those comments are valuable and very helpful. We have read through comments carefully and have made corrections. The responses to your comments are presented following.
Point 1: The Authors need to recheck the writing of the gene name in the title - PKS. I think it should be in italics.
Response 1: Thank you for this suggestion. PKS represents polyketide synthase. While our study is focused on the knockout of pigment genes of T. hamatum T21. Thus, according to your suggestions, we have corrected the “PKS” into “Thpks1”.
Point 2: Correct how to write a term - CRISPR/Cas9 or CRISPR-Cas9 (Line 57 and several other instances). Following a uniform way of writing the text throughout the draft would be ideal.
Response 2: Thanks for your careful checks. We are sorry for our carelessness. Based on your comments, we have made the corrections to make the word harmonized in the whole manuscript.
Point 3: Line 158-170 – sgRNA (in fact, spacer) sequences generally are without the PAM nucleotides, but authors have included them in spacers. Also, the term- sgRNA or gRNA sequence is named for the combination of both the spacer (crRNA) and scaffold (tracrRNA) region (refer to this - https://doi.org/10.1007/978-981-16-9720-3_10). Revise the text to include the correct information.
Response 3: We sincerely thank the reviewer for careful reading. According to your suggestions, we have completed the modification in the text by consulting the literature.
Point 4: Figure 2- Labels added inside the panel can be better reorganized.
Response 4: We have reorganized this part according to your suggestion.
Point 5: Figure 4- cas9, not cas9; HR arm labeling in the gene disruption- up and down or up and up?
Response 5: It is our negligence and we are sorry about this. As suggested by the reviewer, related content has been corrected.
Point 6: As mentioned by the authors, in previous studies, the knockout experiments for the pyr4 and pks1 genes were unsuccessful, possibly due to different DNA repair modes in other strains. What is the author's opinion about succeeding the KO of these genes in the tested strain? This aspect should be discussed more clearly in the discussion section.
Response 6: We think this is an excellent suggestion. We have added corresponding content in the discussion section. In our previous study, neither homologous recombination nor split marker methods could successfully knock out the genes of T. hamatum T21. The main reason for successful knockout is that the CRISPR/Cas9 system can cause double-strand breaks (DSBs). DSBs can be repaired by NHEJ or HR to create genomic alterations, gene insertions, and gene knockouts. In addition, Thpyr4 could be utilized as a bidirectional selection marker to resolve the antibiotic screening markers deficiency and achieve continuous multigene knockout in the fungus. Up-to-date, the biosynthesis of green pigments is a complex process in Trichoderma. Further study of biosynthetic pathways and regulatory mechanisms using the CRISPR/Cas9 system. The above content has been described in the discussion section.
Point 7: The authors should differentiate the data and their interpretation of HR and gene knockouts generated for targeted genes.
Response 7: We sincerely appreciate the valuable comments. In this study, after screening by G418, 242 transformants were acquired, and 175 of them showed green fluorescence under the fluorescence microscope. Based on phenotype and molecular identification, 156 transformers were successfully knockout with Thpks1, of which 27 knockout mutants were due to Cas9 nuclease shearing and GFP inserts randomly in the genome. However, 129 knockout mutants with successful insertion of the GFP at the dual target site, achieved homologous recombination. In previous studies, the author could not obtain Thpks1 knockout mutant by homology recombination and split-marker methods in T. hamatum. Compared with previous studies, indicating that the efficiency of homologous recombination increased strikingly with CRISPR/Cas9-mediated editing, with a knockout efficiency of 73.7%. This part of the analysis is described in the Results and Discussion section.
Special thanks to you for your good comments.